# Characterization of Date Seed Powder Derived Porous Graphene Oxide and Its Application as an Environmental Functional Material to Remove Dye from Aqueous Solutions

**DOI:** 10.3390/ma15228136

**Published:** 2022-11-16

**Authors:** Fatimah A. M. Al-Zahrani, Badria M. Al-Shehri, Reda M. El-Shishtawy, Nasser S. Awwad, Khalid Ali Khan, M. A. Sayed, Saifeldin M. Siddeeg

**Affiliations:** 1Chemistry Department, Faculty of Science, King Khalid University, P.O. Box 9004, Abha 61413, Saudi Arabia; 2Research Center for Advanced Materials Science (RCAMS), King Khalid University, P.O. Box 9004, Abha 61413, Saudi Arabia; 3Unit of Bee Research and Honey Production, Faculty of Science, King Khalid University, P.O. Box 9004, Abha 61413, Saudi Arabia; 4Chemistry Department, Faculty of Science, King Abdulaziz University, P.O. Box 80203, Jeddah 21589, Saudi Arabia; 5National Research Centre, Dyeing, Printing and Textile Auxiliaries Department, Textile Research Division, Dokki, Cairo 12622, Egypt; 6Applied College, King Khalid University, P.O. Box 9004, Abha 61413, Saudi Arabia; 7Physics Department, Faculty of Science, King Khalid University, P.O. Box 9004, Abha 61413, Saudi Arabia; 8Physics Department, Faculty of Science, Al-Azhar University, P.O. Box 71452, Assiut 71524, Egypt

**Keywords:** phenothiazine, adsorption, water-insoluble dye, GO-date seeds

## Abstract

This study aims to prepare graphene oxide (GO) from raw date seeds (RDSs), considered one of the available agricultural wastes in Saudi Arabia. The preparation method is done by the conversion of date seeds to lignin and then to graphite which is used in a modified Hummer’s method to obtain GO. The adsorption of insoluble phenothiazine-derived dye (PTZS) over raw date Seeds (RDSs) as a low-cost adsorbent was investigated in this study. X-ray diffraction (XRD), scanning electron microscopy (SEM), and Fourier-transform infrared spectroscopy (FTIR) were used to characterize (RDSs). According to the calculations, Freundlich isotherms and pseudo-second-order accurately predicted the kinetic rate of adsorption. The adsorption ability was 4.889 mg/g, and the removal rate was 93.98% GO-date Seeds mass, 11 mg/L starting dye concentration, at a temperature of 328 K, pH 9, and contact length of 30 min by boosting the PTZS solution’s ionic strength. In addition, the computed free energies revealed that the adsorption process was physical. Thermodynamic calculations revealed that dye adsorption onto GO-date seeds was exothermic and spontaneous.

## 1. Introduction

Dye chemicals are among the most environmentally harmful organic compounds [1,2]. More than 40,000 different dyes and pigments are enumerated by Chakrabarti et al. [3], and around 12% of synthetic dyes are lost during production and processing procedures. Approximately 20% of these colors are in industrial wastewater [4,5]. Although various technologies for removing contaminants from wastewater exist, such as chemical oxidation, coagulation, electrochemistry and membrane separation processes, and aerobic and anaerobic biodegradation, these treatments are ineffective due to various factors [6]. Adsorption techniques can efficiently and effectively remove organic dyes [7,8,9,10,11,12]. Wastewater treatment technologies using adsorption have a positive economic impact [13,14,15].

The adsorbent materials should be small because large surface areas produce high adsorption efficiency. Aqueous waste streams loaded with hazardous materials would be difficult to separate from water due to the small size of the particles. Various adsorbent materials were investigated to eliminate organic dyes from waste streams. Specifically, the application of activated carbon [16] has long been the norm [17,18]. Many adsorbents have been developed in recent years [9] for the removal of dyes, such as layered double hydroxides [19], metal oxides, clay [20,21,22], natural goethite [23], and sediments [24,25].

Alternative adsorbents for dyes have been proposed using bio-adsorbent materials. Raw date seeds (RDSs), in particular, have gotten much press because of their low cost, natural availability, and low environmental impact. The date palm’s fruit comprises a seed and fleshy pericarp. Date powders, pitted dates, date syrup, chocolate-coated dates, date juice, and date confectionary are all produced by date fruit processing plants, while date seed that has been pitted is a waste material [26]. Furthermore, the RDS is globally distributed and abundant, making them attractive environmental adsorbents for application in industrial processes [3]. To evaluate the elimination of nonyl phenol and bisphenol A from aqueous solutions, they utilized carbonaceous date pits supplemented with nanoparticles of ZnO. They discovered that the maximal removal rate under ideal conditions was 95% [27,28]. Date seeds, primarily composed of cellulose, hemicellulose, and lignin, are effective materials that can be used as an adsorbent to remove organic and inorganic pollutants from aqueous solutions. The success of these low-cost sorbents is primarily due to oxygenated functional groups found in lignocellulosic materials such as cellulose and phenolic compounds [29].

On the other hand, the adsorption of PTZS onto RDS has not been thoroughly investigated. Adsorption is a convenient, easily operated, and designed method. The adsorption process is considered a better alternative in water and wastewater treatment to remove a wide variety of dyes than other methods, such as ion exchange, as mentioned in the study of Bhatnagar et al. [30].

Bukola M Adesanmi et al., (2022) found that the average removal percentage was 98. 23 in adsorption, while the average removal percentage was 90 in ion exchange [31].

As a result, the purpose of this research is to start investigating the physics and chemical characteristics of RDS bio-sorbent that used a range of techniques, including SEM, XRD, and FTIR, and then assess the efficacy of using RDP as an environment-friendly, natural, and cost-effective bio-adsorbent for the separation of PTZS in an aqueous solution.

Furthermore, factors influencing the adsorption of the PTZS method on the RDS bio-adsorbent were solution pH, adsorbent mass, RDS particle size, and PTZS starting concentration. This research aims to provide a simple process for producing sorbent RDS material from a low-cost natural source, namely date seed, a common dietary waste in Saudi Arabia with high local availability and low cost. The RDS sorbent materials were activated to boost the adsorption performance using physical or chemical treatment procedures. Activated sorbent samples were utilized to remove dye from polluted aqueous samples.

## 2. Materials and Methods

### 2.1. General

All reagents and solutions were purchased from Sigma-Aldrich and used as directed. In a DMSO: D_2_O-d_6_ solution, ^1^H and ^13^C NMR spectra were collected using a 400 MHz spectrometer Bruker Advance. Infrared spectra were collected using a 100 FTIR spectrometer of PerkinElmer spectra. Mass spectrometers (Agilent GC 7000) were used for mass spectroscopy. On a UV-VIS Spectrophotometer (Shimadzu, Kyoto, Japan), UV absorption spectra in several solvents were obtained. The PerkinElmer LS 55 Fluorescence Spectrometer was used to record the fluorescence spectra.

### 2.2. Characterization and Synthesis

#### 2.2.1. Synthesis of Organic Dye

(E)-2-cyano-3-(10-octyl-9,10-dihydroanthracen-2-yl) acrylic acid (PTZS) A mixture of 10-Octyl-10H-phenothiazine-3-carbaldehyde (1.18 g, 3 mmol) (1.18 g, 3 mmol), which was prepared as described in the literature [32] and 0.395 g cyanoaceticacid of 6 mmol, diluted in a basic solution of ethanol (7 mL) was agitated overnight then filtered off, and purified at room temperature by chromatography (column) to produce 74.6% (1.4 g). The result of characterization are M.p. 58–59 °C; ^1^H NMR (400 MHz, DMSO:D_2_O-d_6_) δ 3.992 (t, 3H, CH_2_-N), 7.238 (d, 2H, J = 8.8 Hz, Ar-H), 7.657 (d, 2H, J = 2.0 Hz, Ar-H), 7.829 (dd, 2H, J = 8.8, 2.0 Hz, Ar-H), 8.212 (s, 2H, C=C-H). ^13^C NMR (125 MHz, DMSO:D2O-d_6_) δ 14.30, 22.82, 26.96, 27.03, 29.35, 29.39, 31.92, 48.23, 115.01, 116.16, 123.78, 124.04, 125.25, 127.76, 127.78, 128.64, 130.27, 131.26, 143.67, 150.99, 190.26. The chemical structure of PTZS dye is mentioned in Figure 1.

#### 2.2.2. Synthesis of GO-Date Seeds Adsorbent

A total of 100 g of date seed powder was combined with 140 cc of 98% sulfuric acid in this experiment. For a while, the mixture was left to sit. The combination was cleaned after 2 h in a conical flask and diluted with 3% sulfuric acid. Under reflux, the mixture was heated for 4 h. The residue was washed and dried at 105 °C in an oven, chilled, and quantified as acid-insoluble lignin [33].

As in the previous section, this study used lignin from date seeds as a carbon source. In this study, the catalyst source was iron (III) nitrate. Using a co-precipitation approach at room temperature, an iron-lignin precursor with 10% iron loading was made. To make the lignin solution, 35 g of lignin was mixed with 50 mL of tetrahydrofuran (THF) for two hours. Meanwhile, iron (III) nitrate nonahydrate (41 g) was added to a glass beaker containing 100 mL distilled water and stirred until the metal salt was dissolved, resulting in an iron nitrate solution. The iron nitrate solution was poured into the lignin solution at 70 °C and agitated for 2 h. The iron-lignin combination was heated and dried for 24 h at 150 °C after 1 d at room temperature. A solid iron-promoted lignin combination was formed as a result. A quartz tube reactor was then loaded with the iron-promoted lignin precursor. The reactor was heated to 900 °C each minute at a pace of 10 °C, N_2_ gas was released, and the CO_2_ gas was then passed at a 10° per minute pace. The furnace was automatically cooled at a rate of 10 °C/min. A few layers of graphite were retrieved from the quartz tubular reactor for graphite oxide conversion.

Hummer’s method was modified to make GO by oxidizing graphite and removing the catalyst from the graphite. In a 2000 mL flask, we combined 15 g of graphite with 200 mL, 98 percent H_2_SO_4_, and 5.5 g NaNO_3_, then cooled the liquid to 0 °C in an ice bath and shook it for 10 min. Then, for the next 30 min, 33 g of KMnO_4_ was gradually added while stirring continuously. After that, the mixture was diluted with 1 L of distilled water and then heated at 75 °C for 20 min. After that, the combination was treated with 500 mL of 10% H_2_O_2_ to stop the process. The reaction time from graphite to graphene oxide was one hour, far faster than graphene oxide generated from natural graphite. After centrifugation and sonication, the GO solution was rinsed. After the freeze-drying process, 1.0 g of GO-date powder was produced from the GO solution [33].

### 2.3. Adsorption Study

All adsorption batches were run into glass Teflon-capped bottles, which were shaken by a water bath orbital shaker at 100 rpm under constant pressure and temperature conditions. Typically, the fixed amount (mg) of adsorbent was placed with a fixed volume (L) of adsorbate solution (A1) at an initial concentration (C_o_ mg/L). The organic dye solution was obtained by preparing a stock solution of 1 g/L in double distilled water.

The total concentration of dye in the solution at the initial time (C_o_) or at (t) time (C_t_) by (mg/L) was detected via a double beam UV-Vis spectrophotometer (Shimadzu, 1800, Kyoto, Japan) between 300 nm and 800 nm at applied lambda max (445 nm) PTZS dye according to Beer-Lambert Low [34] for the test in which we will remove it. By varying the concentrations of NaOH and HCL reagents, the PH impact was examined as a typical choice for controlling of PH effect of adsorption tests in the previous literature. The solution pH was detected by using a Thermo Scientific pH meter.

The kinetics of the removal of the filtering of the samples revealed information about adsorbate. After the desired contact time, the filtrate was analyzed for the remaining adsorbate concentration. The same procedure was used to investigate the impact of contact time (20–120 min), initial 2-chlorophenol concentrations (40–200 mg/L), and adsorbent dosage (0.1–0.7 g/L) pH (2–12) and temperature (2 g per 50 mL solution) (20–90 °C). All adsorption batches were run into glass Teflon-capped bottles, which were shaken by a water bath orbital shaker at 100 rpm under constant pressure and temperature conditions. Typically, 100 mg of adsorbent was placed with 25 mL of dye solution, and the suspension was filtered after a predetermined time interval. At the same time, the initial concentration of dye at the initial time, which was coded (C_o_) and at (t) time coded (C_t_) by (mg/L), was detected via a double beam UV-Vis spectrophotometer (Shimadzu, 1800, Kyoto, Japan) between 300 nm and 800 nm at applied lambda max (445 nm) PTZS dye according to Beer-Lambert Low. The kinetics of the adsorption test exhibited preliminary evidence that the required time to achieve equilibrium conditions (contact time) was 30 min in all runs, and the equilibrium dye concentration (C_e_) was measured in (mg/L).

The following equation was used to compute the adsorption percentage [35]
(C_o_ − C_e_)/C_o_ × 100 = %R(1)

Experimental data were expressed as a relationship between equilibrium dye concentration (C_e_ mg/L) and dye adsorption capacity q_t_ (mg/g) at the given time (t) and temperature.

The following correlation was used to compute the adsorption capacity [36]
Qt = (C_o_ − C_t_)V/m(2)
where (V) is the dye solution volume in liters, and m is the mass of the adsorbent (mg). The same procedure was used to investigate the impact of contact time (5–75 min), initial concentrations (1.2–28 mg/L), adsorbent dosage (0.1–0.7 g/L) pH (2–11), and temperature (25–65 °C). A blank test of dye adsorption was also checked without the addition of an adsorbent for investigated adsorbent effect.

## 3. Results and Discussion

### 3.1. Synthesis Dye

The synthesis dye was made as described in the previous study [32]. The dye was obtained by condensing 10-Octyl-10H-phenothiazine-3-carbaldehyde with a high yield of cyanoacetic acid via the Knoevenagel process. IR, NMR, and high-resolution mass spectra were used to confirm the chemical structure of the synthesized molecule (see Appendix A).

### 3.2. Characterization of GO-Date Seeds Powder

XRD analyzed the crystalline structure of prepared GO, GO-date Seeds, and the graphite in the 2θ = 10–80°, as shown in Figure 2. The XRD results of the GO-date Seeds show a broad peak between the angle (2θ) of 20° to 33°. In addition, it lacks a horizontal primary line. Those display that the majority of materials have an amorphous nature [34]. The peaks recorded at (100) 42.53°, (101) 45.68°, and (004) 53.18° demonstrate the crystalline graphite structure. The XRD pattern of graphite displays a distinguishable peak (002) at 26.5° with an inter-planar distance, d 002, of 0.334 nm. This suggests that graphite is a carbon substance with a high degree of orientation. On the other hand, the XRD pattern of GO shifted from 26.5° to 11.66°, corresponding to an inter-planar distance of 0.80 nm. The rise in the inter-planar distance of GO is attributed to the introduction of different functional groups of oxygen during the oxidation process [37].

The conjugated and carbon-carbon double bonds, which result in significant amplitude peaks in the Raman spectrum, can be evaluated using the Raman spectra of the generated GO of GO-date Seeds (Figure 3). A G band describes the typical Raman spectrum of GO at 1605 cm^−1^, which corresponds to the E_2_g phonon of a D band at 1353 cm^−1^, and sp^2^ C atoms, which correspond to the breathing mode of k point phonons of A1g symmetry.

Figure 4 displays GO-date Seeds, graphite, and graphene oxide SEM images. There were significant differences between them in surface morphology. These images of GO-date Seeds showed a smooth surface and absence of pores [38]. On the other hand, the SEM images of GO exhibited porosity highly comparable to GO-date Seeds. Pores formed on the surface of GO are sites for dyes for uptake onto its surface. The SEM micrographs of the GO present that the GO possesses a sheet-like structure. The borders of separate sheets may be recognized from the SEM pictures, which clearly show that GO has numerous lamellar layer structures and wrinkled areas [39].

The functional groups of the GO-date Seeds and GO was investigated using an FTIR spectrometer. Figure 5 depicts the spectra acquired in this study. Because DS contains hemicellulose and cellulose in its fiber, the GO-date Seed spectra reveal a large hydroxyl group (O-H) signal at 3339.9 cm^−1^, possibly due to water molecules. The carbonyl group (C=O) adsorption band was present at 1688.5 cm^−1^, and its stretch indicated the presence of oxygen.

Furthermore, the current peak at 1524.5 cm^−1^ coincided with the C=C double bond stretching, which indicated that alkenes, alkynes, or aromatic group chemicals were also found in the DS C–OH deformation and C–O straining of phenolics, are responsible for the band at 1282 cm^−1^. The band describes the C–O stretching vibration of cellulose and hemicellulose at 1066 cm^−1^. The cellulose vibrations of C–H rocking are responsible for the absorption at 872 cm^−1^ [34]. In the spectrum of FTIR of GO, because of extensive oxidation, GO has a broader O–H stretching vibration band at 3339.9 cm^−1^ (carboxylic group), carboxyl C=O stretching band in GO shifted to 1688 cm^−1^, C=C vibration band shifted to 1651 cm^−1^, the C–OH vibration band shifted to 1136 cm^−1^, and C–O stretching vibration at 1066 cm^−1^. Also, the alkanes group (C–H) peak stretched at 2922.2 cm^−1^ and at 2855.1 cm^−1^ vanished. According to the FTIR analysis, the produced GO-date Seeds include oxygen-containing functional groups [40].

An N_2_-BET surface area study investigated materials’ textural qualities. The surface area of GO-date Seeds ranges from 1.65 m^2^/g [41] to 80.49 m^2^/g [42], while our data were measured at 28.70 m^2^/g. The average pore volume, pore diameter, and specific surface area of GO were measured at 0.073 cm^3^/g, 2015.64 m^2^/g, and 2.68 nm, respectively. The surface area of an adsorbent determines its physical adsorption capacity (see Figure 6). The surface area of GO-date Seeds is large enough to interact with dyes, as evidenced by N_2_-BET surface area data [43,44].

### 3.3. The Adsorption Performance of GO-Date Seeds

#### 3.3.1. Adsorption Performance of GO-Date-Seeds

The adsorption capacity of GO-date seeds towards PTZS dye removal was investigated. A value of 0.05 g of GO-date seeds distributed in 25 mL of PTZS dye solution (C_0_ = 27 ppm) with shaking at 298 K and pH = 9 had a 98% adsorption capacity. The dye solution’s reminder concentration was determined after 60 min.

#### 3.3.2. The Impact of the Initial pH

The sample’s pH is a key component in controlling adsorbent surface protonation. Adsorption tests were performed using 0.05 g graphene oxide dispersed in 25 mL PTZS dye solution (C_0_ = 11 ppm) for 30 min at 298 K with shaking, and the adsorption percentage (Ads. percent) Figure 7 shows the pH of the dye solution as a function of time. The utmost (Ads. %) was 67.9%, 89.2%, 87.5%, 98.98%, and 72.6% at a pH range of 2, 4, 7, 9, and 11, respectively. At pH = 9, the adsorption capacity was nearly 100% of pollutant removal [45].

#### 3.3.3. The Impact of Contact Time

The time dependence of PTZS dye adsorption on GO-date Seeds was evaluated to determine the contact time for adsorption as shown in Figure 8 as a function of time when 0.05 g of graphene oxide adsorbent was dispersed in 25 mL of PTZS dye solution (C_0_ = 11 ppm) at pH 9 and 298 K up to 75 min with constant shaking. The adsorption capacity was achieved in two steps: A relatively rapid step (steep slope) that lasted up to 30 min, followed by a slower step that took longer to reach equilibrium. The enhanced initial adsorption capability is due to the availability of a more significant number of active sites for PTZS dye adsorption at the start of the process. Furthermore, the textural features of the graphene oxide adsorbent allow the PTZS dye to pass into the pores (intra-particle diffusion) and initiate the adsorption process at the active sites on the pore surfaces.

#### 3.3.4. The Effect of Initial Concentration

To better understand the effect of the starting concentration of PTZS dye on the adsorption ability of GO-date Seed adsorbent, the investigations were carried out with different initial concentrations of PTZS dye solution. For this research, 50 mg graphene oxide adsorbent was added to a 25 mL dye solution with an initial concentration of 1.2, 4.1, 12, 27, or 28 ppm at pH = 9 at 298 K for 30 min while shaking. Figure 9 shows the acquired data, which presented the decrease in percent adsorption with the dye concentration increase. Furthermore, the highest adsorption capacity was obtained at 1.2 mg/L and fell as the initial dye concentration was raised, with the maximum adsorption around 98%. The increase in dye concentration led to the saturation of the solution by the dye molecules, which made it difficult for the adsorbent to adsorb the extra adsorbent molecules [39].

### 3.4. Adsorption Kinetics

With PTZS dye adsorption, three well-known kinetics models, namely Elovich models, pseudo-first-order, and pseudo-second-order, were utilized to identify the kinetic parameters. These kinetic models’ linear expressions can be used in the following ways [46]
ln(q_e_ − q_t_)= ln q_e_ − K1 t(3)
t/q_t_ = 1/(K_2_ q^2^) + t/q_e_(4)
q_t_ = β ln(αβ) + β ln t(5)
where (q_e_) and = (q_t_) are the quantities of adsorbate (PTZS) dye adsorbed at equilibrium and at any time t in (mg/g), respectively. K1 and K2 denote the pseudo-first-order (min^−1^) and pseudo-second-order (g/mg. min) rate constants, respectively; and denote the initial adsorption rate (mg/g. min) and adsorption. In addition, because all previous models failed to clarify the diffusion process and phases that determine the rate, the Intra-Particle-Diffusion was used. The model established by Weber and Moris [47] is as follows:q_t_ = K_diff_t^(1⁄2)^ + C(6)
where also (q_t_) as represented previously and (K_diff_) is the rate of intra-particle diffusion constant (mg/g. min 0.5); C is an event that indicates the boundary layer breadth (mg/g).

Figure 10 depicts the fitted data derived from these models’ experimental findings (Figure 10a–c), while the kinetic factors are included in Table 1. The experimental data were better fit by a pseudo-second-order model with an R^2^ close to 1.000. Additionally, the adsorption capacity calculated by this model (4.883 mg/g) was significantly closer to the experimental value (4.920 mg/g) than the pseudo-first-order model’s computation (R^2^ = 0.781). These findings indicate that pseudo-second-order is a good fit for the adsorption process. The Elovich model frequently fits the chemisorption process on an energetically heterogeneous solid. The experimental data for the studied sample did not fit this model (R^2^ = 0.0.496), showing that physisorption is the most likely adsorption mechanism. Figure 9 shows how the Weber-Moris Intra-Particle Diffusion model fits the experimental results. The figure is not linear across the period investigated and does not cross through the source, indicating that adsorption is not a one-step process. Two overlapping lines were visible in the graph, one representing rapid surface adsorption and the other a semi-stable intra-particle diffusion phase (second line).

### 3.5. Adsorption Isotherm Studies

Because temperature affects the process of adsorption, the impact of temperature on adsorption efficiency was investigated using 0.05 g for 25 mL of PTZS dye solution at an initial concentration of 11 ppm and a pH of 9, as shown in Figure 11. According to the findings, the adsorption capacity declined from 53.7% to 4.43% as the temperature climbed from 298 to 338 K. PTZS dye adsorption is an exothermic process and was the first indication of this behavior. This attitude was indicated by the desorption of adsorbate molecules from the adsorbent’s surface and their subsequent migration to the solution. In general, the adsorption capacity declined as the temperature climbed.

Three distinct isotherm models, each with its own set of assumptions, were developed to understand the adsorption process better: Freundlich, Langmuir, and Dubinin-Radshkevich were implemented and coupled with the acquired data. Figure 12a–c shows the fitted equilibrium data acquired from the adsorption isotherm of three prior models, and Table 2 lists the estimated coefficients.

These isotherm models’ linear expressions can be used in the following ways:1/(K_L_q_max_) + Ce/q_max_ = Ce/q(7)
1/n log Ce +logK_F_ = log *q_e_*(8)
In q_(D–R)_ − βε^2^ = ln *q_e_*(9)
RT ln (1 + 1/C_e_) = ε(10)
E = 1/√2β(11)
where q_max_ is the maximum quantity of adsorbate at equilibrium (in mg/g); Ce is the concentration of adsorbate (mg/L); the Langmuir isotherm constant (L/mg), intensity factor, Freundlich constant [mg/g (mg/L) 1/n] are KL, n, and KF, respectively; q(D–R); and, respectively, β and ε are the Polanyi′s potential (mol^2^/kJ^2^), hypothetical adsorption capacity at saturation (mg/g), and adsorption energy for each adsorbate mole (mol^2^/kJ^2^). R is the gas constant (8.314 × 10^−3^ kJ /K mol) (K), and T is the applied temperature. Even when an adsorbate particle travels from infinity to the adsorbent’s solid surface (kJ/mol), E is the average energy per adsorbate molecule.

The more substantial correlation coefficient (R^2^ ≈ 0.96) was well fitted to the Freundlich isotherm model of the PTZS dye adsorption, an improvement on the model of Langmuir. These data support the physisorption hypothesis in which the adsorbate adsorption is multi-layered on the adsorbent surface.

Figure 12c shows the model of D–R isotherm, which detects (q_(D–R)) and (β)from the intercept and slope of the ln q_e_ vs. ε^2^ plot, which is followed by the calculation of free energy using the equation:1⁄(√(2β) = E

If the E value is more significant than 8 KJ/mol, the adsorption is chemisorption in nature; if it is less than 8 KJ/mole, the adsorption is physisorption in nature. The essential issue is that this model may exhibit implausible behavior in the presence of a high adsorbate concentration. Because the computed free energies were substantially higher than 8 KJ/mole, the PTZS was determined to be physical adsorption.

### 3.6. Thermodynamic Study

The thermodynamic portion was regarded as a critical section for calculating G, H, and S factors, where the effect of temperature was explored by utilizing GO-date Seeds in a solution of aqueous at pH = 7 and temperatures of 5 different levels (25, 35, 45, 55, and 65 °C).

The expressions for these thermodynamic parameters were used as follows at first.
q_e_/C_e_ = K_d_
for equilibrium constant (10)
−RT ln K_d_ = ∆G 
for Gibbs energy change (11)
∆S/R − ∆H/RT = In K_d_
for entropy and enthalpy change (12).

Where T is the temperature (kelvin). The R is the common gas constant (8.314 × 10^−3^ KJK^−1^mol^−1^); in Figure 11, the values of (∆H/R) and (∆S/R) were plotted, which were computed from the intercept and slope of the line, which is straight in the plot of ln K_d_ vs. 1/T.

The exothermic character of low affinity and adsorption of the adsorbent is related to the significant negative values of H and S (Table 3). The nonspontaneous nature is increased by increasing temperature, with the degree of spontaneity decreasing with increasing temperature. By comparing our results with previous studies, we found the adsorption performance of date seeds gave high removal of water pollution than the other materials (Table 4).

### 3.7. The Reusable of GO-Date Seeds in Adsorption

Graphene oxide is a renewable adsorbent, allowing us to use it for repeated adsorption of pollutants after regeneration. The adsorbed PTZ dye on the surface of GO-date seeds was released. The adsorbent was washed with DI water and immediately used in the next adsorption experiment after becoming dry. After four adsorption cycles (as shown in Figure 13, the regenerated GO retained 90% of its capacity, indicating that GO-date is an excellent reusable adsorbent.

## 4. Conclusions

At the end of this essay, we may conclude that date seeds obtained from food waste in Saudi Arabia were successfully converted into a chemically activated natural GO adsorbent where the adsorption capacity was 98.8% with physical properties and exothermic nature toward the increasing temperature as well as factors affecting the sorption effectiveness of the sorption process. Adsorption kinetic was pseudo-second order. All equilibrium data obtained at different temperatures fit perfectly with the Freundlich isotherm model. The positive values of ΔG and negative values of ΔH and ΔS indicate that the dye’s adsorption process is nonspontaneous and exothermic, and an increase in randomness occurs at the solid or solution interface. The authors were successful in resolving an essential environmental and ecological issue in KSA related to the accumulation of a foremost waste of food; namely date seeds, by recycling and reusing waste to manufacture a naturally activated sorbent material, which was then used to resolve another serious environmental issue, namely water treatment.

## Figures and Tables

**Figure 1 materials-15-08136-f001:**
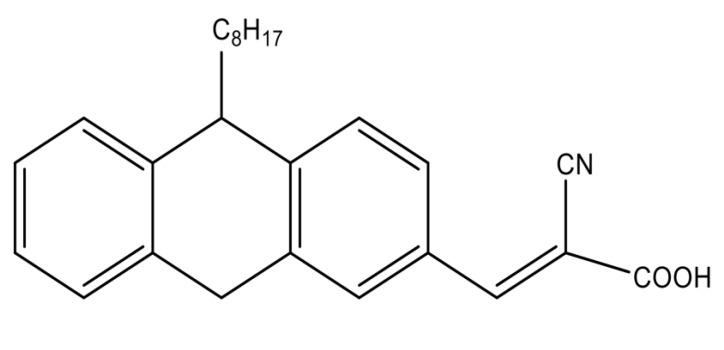
Chemical structure of the organic dye.

**Figure 2 materials-15-08136-f002:**
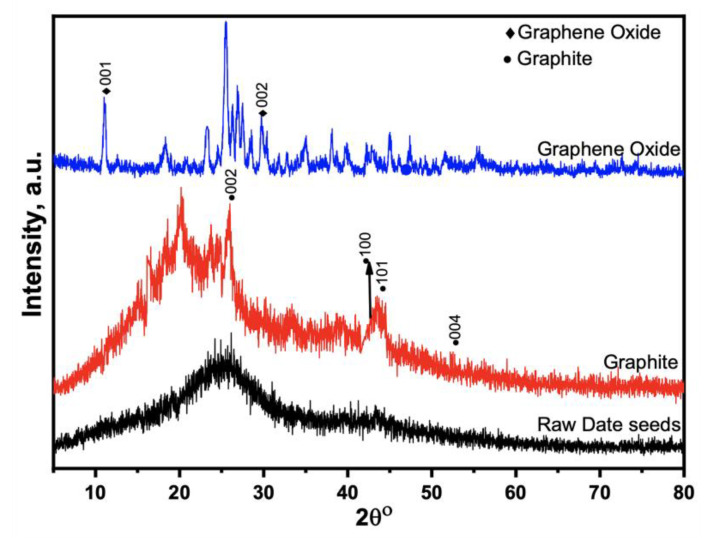
XRD patterns for raw seeds, graphite, and graphene oxide.

**Figure 3 materials-15-08136-f003:**
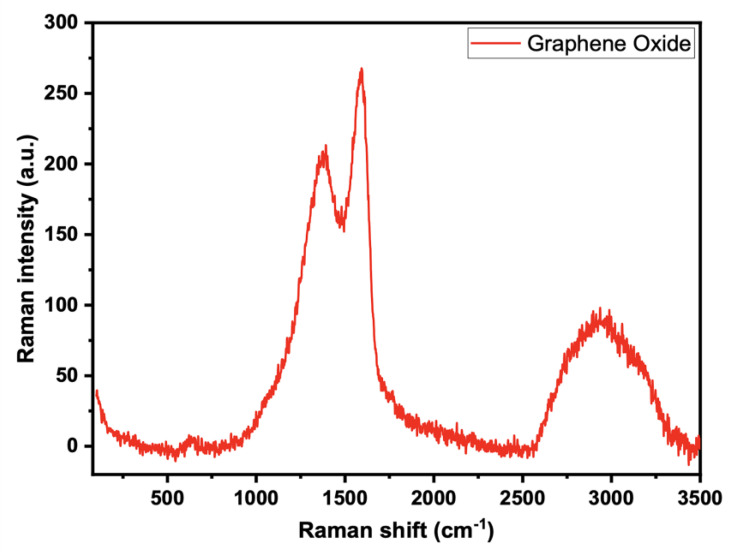
Raman spectra for graphene oxide.

**Figure 4 materials-15-08136-f004:**
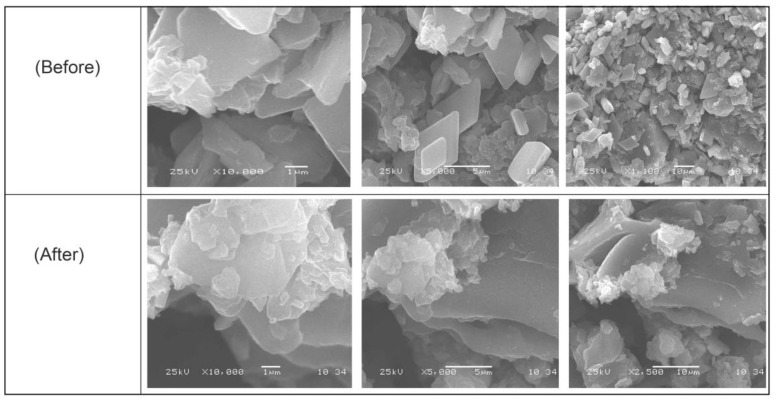
SEM micrograph for GO-date Seeds before and after adsorption.

**Figure 5 materials-15-08136-f005:**
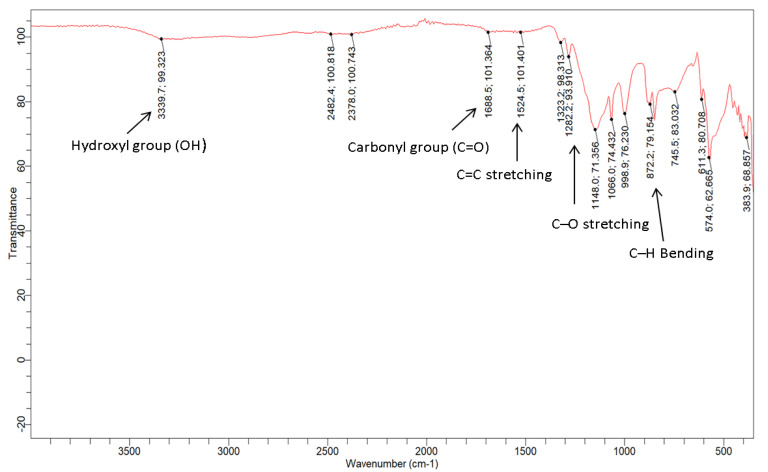
IR-spectra for GO-date seeds.

**Figure 6 materials-15-08136-f006:**
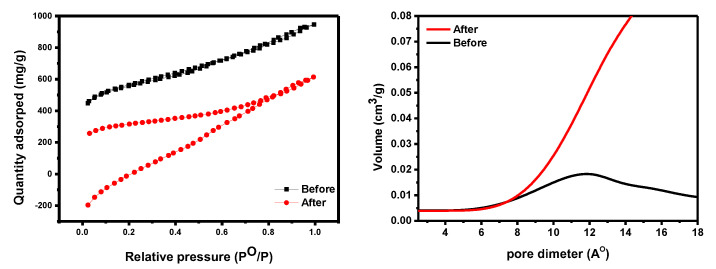
The BET data of GO−date (**left panel**). The pore size distribution of GO−date before and after adsorption (**right panel**).

**Figure 7 materials-15-08136-f007:**
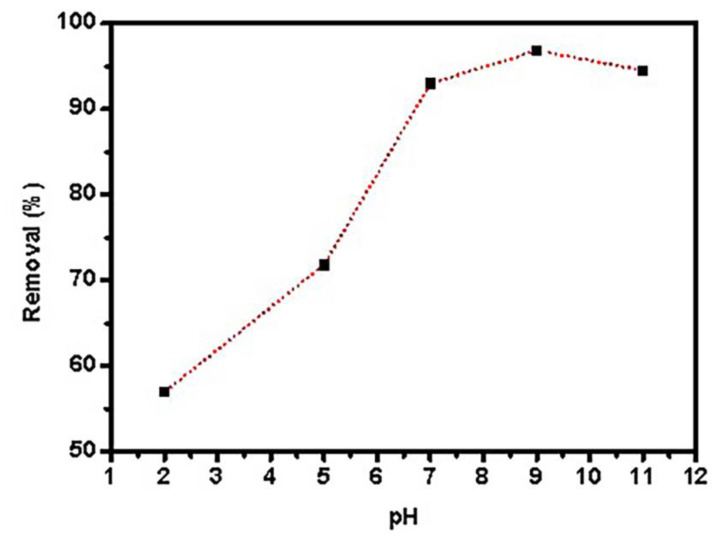
Effects of pH PTZS adsorption by GO-date Seeds (C_0_:11 mg/ L, T: 298 K, Adsorbent dosage: 0.05 g /L, Contact time: 30 min).

**Figure 8 materials-15-08136-f008:**
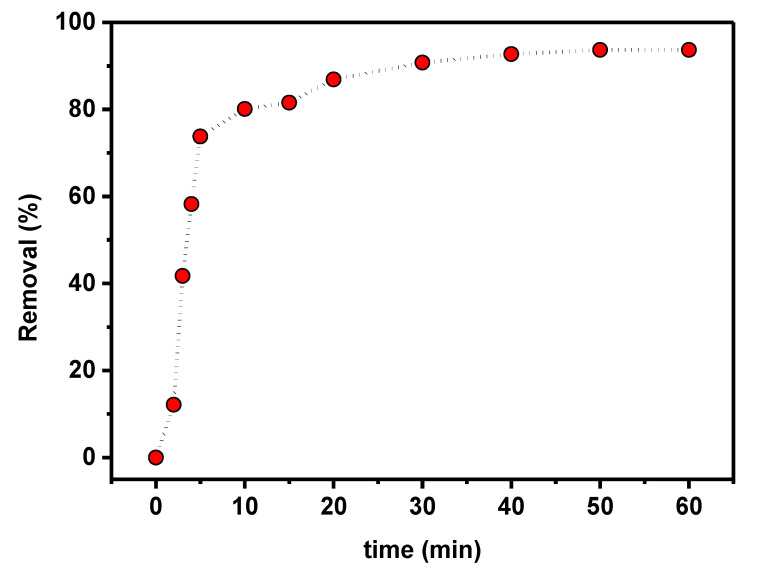
Effect of contact time on PTZS adsorption by GO-date Seeds (C_0_: 11 mg /L, T: 298K, pH: 9, Adsorbent dosage: 0.05 g /L, Contact time: 30 min).

**Figure 9 materials-15-08136-f009:**
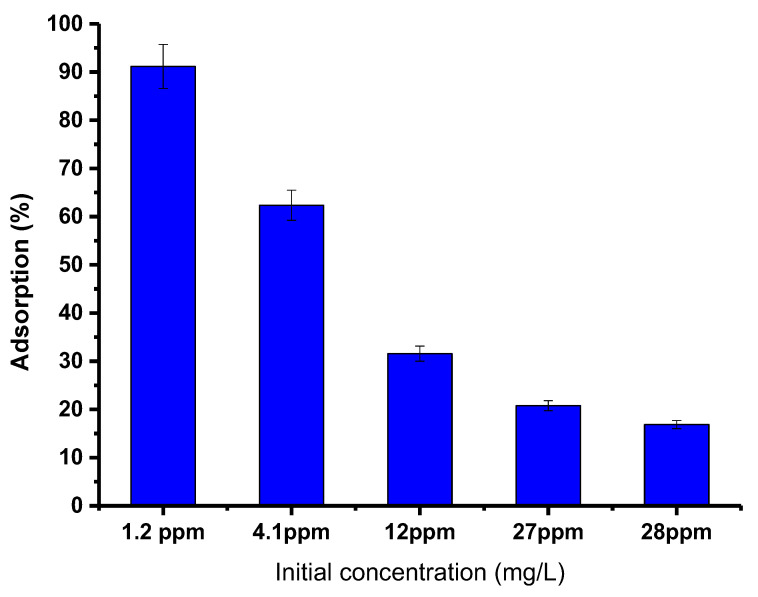
Effect of initial concentration of dye PTZs on adsorption by GO-date Seeds (T: 298K, pH: 9, Adsorbent dosage: 0.05 g/ L Contact time: 30 min).

**Figure 10 materials-15-08136-f010:**
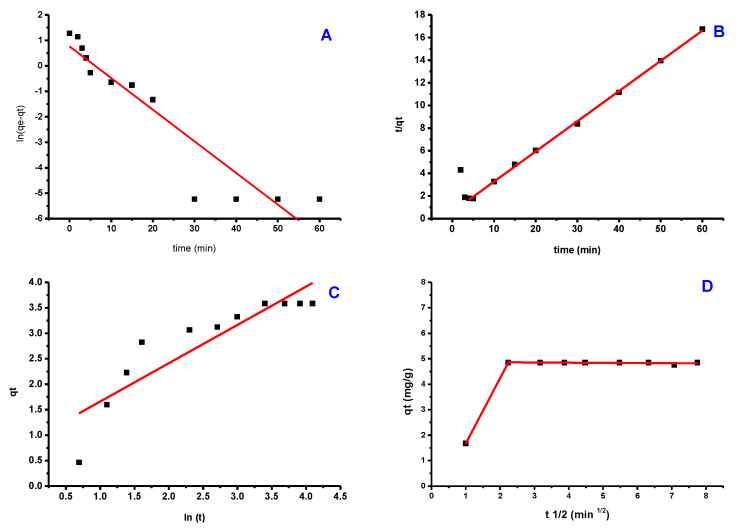
Kinetic models of PTZS dye adsorption GO-date Seeds (**A**) Pseudo−First−Order; (**B**) Pseudo−Second−Order; (**C**) Elovich; (**D**) Weber−Moris Intra−Particle Diffusion Model.

**Figure 11 materials-15-08136-f011:**
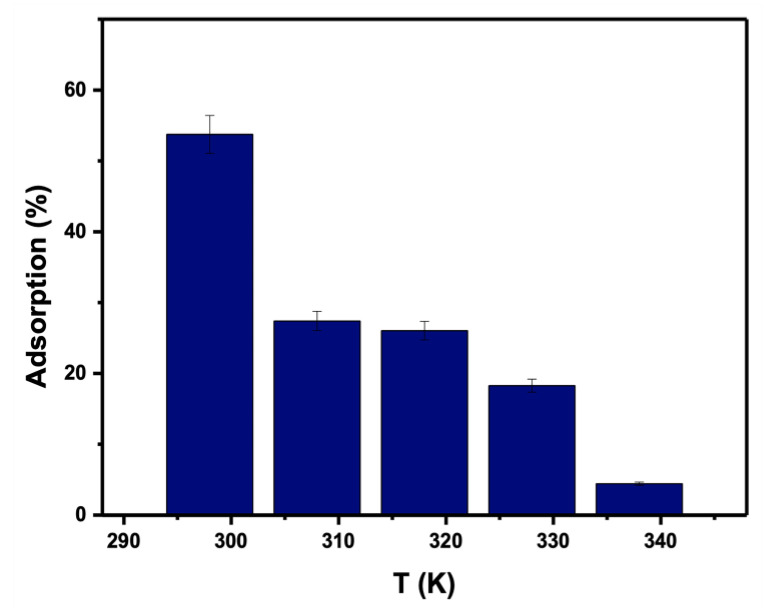
Effect of temperature of dye PTZS on adsorption by GO-date Seeds. (C_o_ = 11 ppm, pH: 9, Adsorbent dosage: 0.05 g /L, Contact time: 30 min).

**Figure 12 materials-15-08136-f012:**
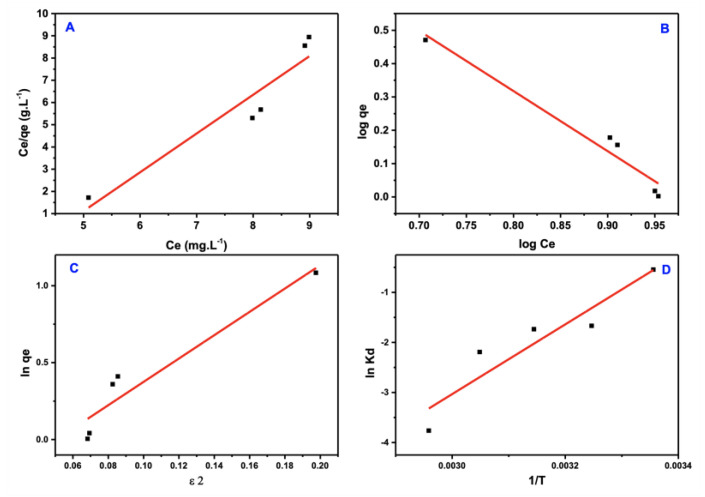
Isotherm models of PTZS dye adsorption. (**A**) Langmuir (**B**) Freundlich (**C**) Dubinin−Ra−dushkevich (D–R) (**D**) Arrhenius equation.

**Figure 13 materials-15-08136-f013:**
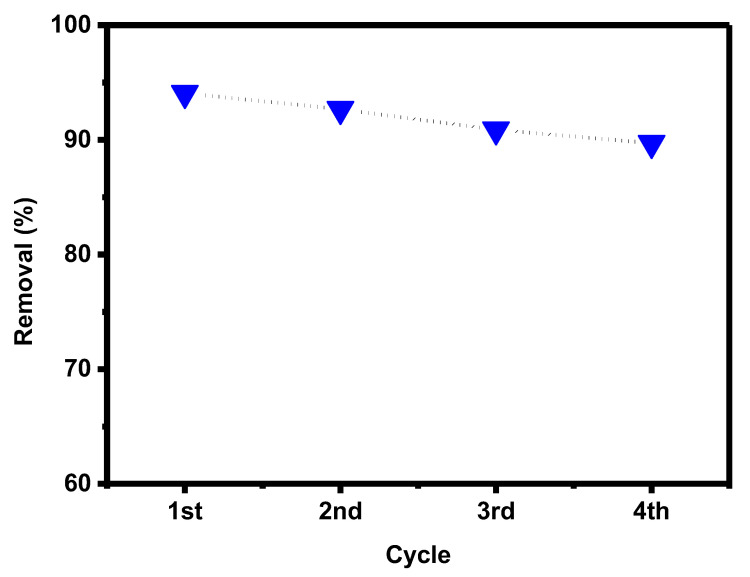
Recycling effect of GO-date Seeds.

**Table 1 materials-15-08136-t001:** Coefficients of Pseudo-First-Order, Pseudo-Second-Order, Elovich kinetic, and Weber-Moris Intra-particle Diffusion models for the adsorption of GO-date Seeds onto PTZS.

**Models**	**Parameters**	**GO-Date Seeds**
	*q* *_e_ Experimentally (mg/g)*	4.920
Pseudo-Frist-Order	q_e_ (mg/g)	0.205
	K_1_ (min^−1^) × 10^3^	21.648
	R^2^	0.267
Pseudo second order	q_e_ (mg/g)	4.883
	K_2_ (g/mg min) × 10^3^	413.295
	R^2^	1.000
Elovich model	α (mg/g min)	3.916
	β (g/mg)	44.977
	R^2^	0.190
Intra-particle Diffusion	K_diff_ (mg/g min)	0.274
	C (mg/g)	4.850
	R^2^	0.344

**Table 2 materials-15-08136-t002:** Comparison of the coefficients of isotherm for PTZS adsorption onto GO-date Seeds.

**Parameters**	**Isotherm Models**					
	**Langmuir Model**		**Freundlich Model**		**D–R**	
	q_max_ (mg/g)	0.57	KF (mg/g.(L/mg)1/n	57.67	q_m_ (mg/g)	0.73
	K_L_ (L/mg)	−0.23	1/n	−1.80	B (mol^2^/J^2^)	7.58
	R_L_	−0.96	R^2^	0.96	E (kJ/mol)	0.26
	R^2^	0.91	-	-	R^2^	0.92

**Table 3 materials-15-08136-t003:** Thermodynamic parameters for adsorption of PTZS dye (25 mg/L) at pH = 9 for 30 min.

t (°C)	T (K)	ln K_d_	∆G(KJ)	∆S(J)	∆H(KJ)
25	298	−0.543	1.346	−144.090	−31.720
35	308	−1.668	4.271		
45	318	−1.737	4.592		
55	328	−2.191	5.975		
65	338	−2.146	6.031		

**Table 4 materials-15-08136-t004:** Adsorption capacities of some adsorbent materials for the removal of dyes from wastewater.

Adsorbents	Adsorption Capacity mg/g	References
Bottom ash	7.4	[48]
Deoiled soya	5.3	[48]
Silica-filled epoxidized natural rubber beads	2.6	[49]
Activated charcoal	0.04	[50]
Barley husk-derived activated carbon	6.67	
GO	4.88	This work

## Data Availability

Not applicable.

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
