# Peer review of "Characterization of Date Seed Powder Derived Porous Graphene Oxide and Its Application as an Environmental Functional Material to Remove Dye from Aqueous Solutions"

_materials, 2022, doi:10.3390/ma15228136_

Round 1

Reviewer 1 Report

Manuscript study deals with the application of raw date Seeds (GO- date Seeds) as a low-cost adsorbent. Furthermore, solution pH, all adsorbent mass, and GO-date Seeds particle size provide essential data on the adsorption procedure, the adsorption approach, the effects of PTZS starting temperature and concentration on concentration on PTZS adsorption were investigated. Equilibrium of adsorption X-ray diffraction (XRD), scanning electron microscopy (SEM), and Fourier-transform infrared spectroscopy (FTIR) is used to characterize GO-date Seeds. According to the calculations, Freundlich isotherms and pseudo-second-order accurately predicted the kinetic rate of adsorption. The surface functional groupings of GO-date Seeds have been significantly influenced by the adsorption properties of PTZS. The adsorption ability was 4.889 mg/g, and the removal rate was 98.98% at 4.88 g/L GO-date Seeds mass, 11 mg/L starting dye concentration, the temperature of 328 K, pH 9, and contact length of 30 minutes by boosting the PTZS solution's ionic strength. In addition, the computed free energies revealed that the adsorption process was physical. Thermodynamic calculations revealed that dye adsorption onto GO-date seeds was exothermic and spontaneous. This article describes a simple approach in Saudi Arabia for recycling significant food waste and environmental pollution to make an activated adsorbent that has been effectively used to remediate contaminated aqueous samples.. After going through the manuscript, I feel that the topic is interesting but the execution and presentation of work needs to highlight the novelty the author should carefully address following comments:

1.     Rewrite abstract it is too verbose present only prominent results.

2.     In introduction section author needs to discuss why their  materials are good for environmental detoxification and why adsorption is better then other methods such as ion exchange and photocatalysis see following works Chemical Engineering Journal 251 (2014), Journal of Hazardous Materials 416,(2021) 125714.

3.     Adsorption studies method need to rewritten in more well-defined way supported with relevant references.

4.     If possible, author should elaborate or add characterization aspect such elemental mapping etc.

5.     To demonstrate stability of their material if possible, please SEM of material before and after adsorption.

6.     Quality of all figures need to be improved.

7.     Rediscuss effect of concentration and adsorbent dosage using see Gels 2022, 8, 23. doi.org/10.3390/gels8010023

8.     Please highlight which adsorption isotherms fits well and what is nature of adsorption chemical or physical.

9.     What functional groups are present on material for adsorption please highlight in text see Carbohydrate Polymers 241(2020), 116258.

10.  Present only prominent results in conclusion section.

11.  Please check all the results mentioned in text and abstract once again.

12.  Compare your adsorption results with other reported works.

Author Response

Ms. No. materials-1990728

Title of the article:

Characterization of date palm seed powder and its application as an environmental functional material to remove dye from aqueous solutions.

RESPONSE LETTER

Reviewer #1:

Manuscript study deals with the application of raw date Seeds (GO- date Seeds) as a low-cost adsorbent. Furthermore, solution pH, all adsorbent mass, and GO-date Seeds particle size provide essential data on the adsorption procedure, the adsorption approach, the effects of PTZS starting temperature and concentration on concentration on PTZS adsorption were investigated. Equilibrium of adsorption X-ray diffraction (XRD), scanning electron microscopy (SEM), and Fourier-transform infrared spectroscopy (FTIR) is used to characterize GO-date Seeds. According to the calculations, Freundlich isotherms and pseudo-second-order accurately predicted the kinetic rate of adsorption. The surface functional groupings of GO-date Seeds have been significantly influenced by the adsorption properties of PTZS. The adsorption ability was 4.889 mg/g, and the removal rate was 98.98% at 4.88 g/L GO-date Seeds mass, 11 mg/L starting dye concentration, the temperature of 328 K, pH 9, and contact length of 30 minutes by boosting the PTZS solution's ionic strength. In addition, the computed free energies revealed that the adsorption process was physical. Thermodynamic calculations revealed that dye adsorption onto GO-date seeds was exothermic and spontaneous. This article describes a simple approach in Saudi Arabia for recycling significant food waste and environmental pollution to make an activated adsorbent that has been effectively used to remediate contaminated aqueous samples. After going through the manuscript, I feel that the topic is interesting but the execution and presentation of work needs to highlight the novelty the author should carefully address following comments.

COMMENT

RESPONSE

1. Rewrite abstract it is too verbose present only prominent results.

The abstract is re-written.

2. In introduction section author needs to discuss why their materials are good for environmental detoxification and why adsorption is better than other methods such as ion exchange and photocatalysis see following works Chemical Engineering Journal 251 (2014), Journal of Hazardous Materials 416, (2021) 125714.

The authors appreciate the positive comment. The date seeds are primarily composed of cellulose, hemicellulose, and lignin, are an effective material that can be used as an adsorbent to remove both organic and inorganic pollutants from aqueous solutions. The success of these low-cost sorbents is primarily due to oxygenated functional groups found in lignocellulosic materials such as cellulose and phenolic compounds.

Further its convenience, ease of operation, and simplicity of design, the adsorption process is considered a better alternative in water and wastewater treatment to remove a wide variety of dyes more than the other method such as ion exchange as the mentioned in this report.  All needful are incorporated in revised manuscript along with references.

3. Adsorption studies method need to rewritten in more well-defined way supported with relevant references.

Adsorption studies method re-written with more defined way supported by some new relevant references.

4. If possible, author should elaborate or add characterization aspect such elemental mapping etc.

We don’t have facility of elemental mapping. It’s not possible to add. 

5. To demonstrate stability of their material, if possible, please SEM of material before and after adsorption.

Figure 4 was revised for SEM before and after adsorption.

6. Quality of all figures need to be improved

All figures are improved with 300 dpi.

7. Rediscuss effect of concentration and adsorbent dosage using see Gels 2022, 8, 23. doi.org/10.3390/gels8010023

Discussion improved with some new references. 

8. Please highlight which adsorption isotherms fits well and what is nature of adsorption chemical or physical.

Adsorption isotherms fits well to Freundlich isotherm model and physical adsorption as mentioned in page 12 line 360 and page 13 line 375.

9. What functional groups are present on material for adsorption please highlight in text see Carbohydrate Polymers 241(2020), 116258

We fitted the results of FTIR as mentioned on page 6, line 209 and compared it with suggested reference.

10. Present only prominent results in conclusion section.

Needful done according to the suggestion.

11. Please check all the results mentioned in text and abstract once again.

Done. 

12. Compare your adsorption results with other reported works.

A new table 4 added that compare the adsorption results in this work along with other reported work

We have also made some other changes in manuscript, e.g., formatting style, references according to your journal requirements.

We appreciate earnestly for Editors/Reviewers’ warm work and hope that the correction will meet with approval criteria.

Thank you very much again for your comments and suggestions.

Yours sincerely,

Dr. Khalid Ali Khan

On behalf of all authors.

Reviewer 2 Report

The authors prepared an adsorbent from date seeds for dye adsorption. However, the data are not presented clearly and show erroneous data, that is, the statements differ from what is seen in the data figures. The study lacks proper characterization and incomplete data is presented. Application wise, the experiments carried out are lacking to provide sufficient evidence that the material is useful for waste water treatment.

 Comments:

1.       What is the “Equilibrium of adsorption X-ray diffraction” mentioned in the abstract?

2.       The XRD pattern presented in Figure 2 is of poor quality with very low signal-to-noise ratio. Kindly replace with a better measurement and also place the calculated value of graphene oxide and graphite for a better comparison.

3.       Mentioned in page 6 “Fig. 4 displays GO-date Seeds, graphite and graphene oxide SEM images.” However, the images were not clearly labeled in the figure (which is graphite? Graphene oxide?). What is the PTZS label in Figure 4? Also, provide similar magnification of samples for better comparison.

4.       Mentioned in page 6 “The adsorption rate of the GO-date Seeds and GO was investigated using an FTIR spectrometer.” How was the adsorption rate measured by FTIR?

5.       Please improve labels on Figure 5. Include functional groups represented by the peaks so it is easy for readers to interpret in one look.

6.       Kindly present the N2 adsorption data and the pore size distribution mentioned in Page 7.

7.       The dye is acidic in nature and placing it in a basic media gives a tendency for the dye to be unstable and breakdown. Kindly present the blank determination or stability tests whereby the dye is exposed to varying pH and subsequently measured for UV-Vis spectra.

8.       Mentioned in page 9 “Furthermore, the adsorption capacity (Ads. per cent) increased as the initial dye concentration was raised…” However, it can be seen from Figure 8 that the adsorption percent lowers as the initial concentration was raised.

9.       Is the adsorption process reversible? Is the material reusable?

10.   How does the current material compare to other materials for waste water treatment available in the literature in terms of adsorption capacity? Kindly tabulate for a better comparison.

11.   Polluted water always contains other contaminants. Kindly present the adsorption capacity of the material in the presence of other contaminants.

12.       Kindly define the RDS abbreviation the first time it is used in the manuscript.

13.       The manuscript uses terms like “cooked” and “baked”. Kindly consider using scientific terms to describe the methodology.

14.       Kindly have the manuscript undergo language editing to correct various errors, as well as inconsistencies in tenses and writing point of view.

Author Response

Ms. No. materials-1990728

Title of the article:

Characterization of date palm seed powder and its application as an environmental functional material to remove dye from aqueous solutions.

RESPONSE LETTER

Reviewer #2:

The authors prepared an adsorbent from date seeds for dye adsorption. However, the data are not presented clearly and show erroneous data, that is, the statements differ from what is seen in the data figures. The study lacks proper characterization and incomplete data is presented. Application wise, the experiments carried out are lacking to provide sufficient evidence that the material is useful for waste water treatment.

COMMENT

RESPONSE

1. What is the “Equilibrium of adsorption X-ray diffraction” mentioned in the abstract?

X-ray diffraction (XRD), scanning electron microscopy (SEM), and Fourier-transform infrared spectroscopy (FTIR) are used to characterize GO-date seeds. According to the calculations, Freundlich isotherms and pseudo-second-order accurately predicted the kinetic rate of adsorption.

2. The XRD pattern presented in Figure 2 is of poor quality with very low signal-to-noise ratio. Kindly replace with a better measurement and also place the calculated value of graphene oxide and graphite for a better comparison.

The XRD pattern was generated through software Origin 95, so based on our data file it is difficult to replace. In this experiment we worked on GO-date seeds and peaks for graphite was used only for comparison. 

3. Mentioned in page 6 “Fig. 4 displays GO-date Seeds, graphite and graphene oxide SEM images.” However, the images were not clearly labeled in the figure (which is graphite? Graphene oxide?). What is the PTZS label in Figure 4? Also, provide similar magnification of samples for better comparison.

The revised figure has only GO-date seeds before and after adsorption experiment. Further, the label inside figure 4 is changed to “before” and “after”

4. Mentioned in page 6 “The adsorption rate of the GO-date Seeds and GO was investigated using an FTIR spectrometer.” How was the adsorption rate measured by FTIR?

We changed the text about the adsorption rate measured by FTIR. The functional groups of GO-date seeds and GO were investigated using an FTIR spectrometer.

5. Please improve labels on Figure 5. Include functional groups represented by the peaks so it is easy for readers to interpret in one look.

Figure revised according to the suggestion.

6. Kindly present the N2 adsorption data and the pore size distribution mentioned in Page 7.

Done

7. The dye is acidic in nature and placing it in a basic media gives a tendency for the dye to be unstable and breakdown. Kindly present the blank determination or stability tests whereby the dye is exposed to varying pH and subsequently measured for UV-Vis spectra.

The dye was stable when placing in nature and a basic media. Whereas, the absorbance was semi-identical as shown in the figure S4 in supplementary file.

8.  Mentioned in page 9 “Furthermore, the adsorption capacity (Ads. per cent) increased as the initial dye concentration was raised…” However, it can be seen from Figure 8 that the adsorption percent lowers as the initial concentration was raised.

Thank you for your value comment. We have revised the text which is describing the figure 8 properly.

9. Is the adsorption process reversible? Is the material reusable?

Yes, the adsorption process is reversible and the material is reusable. We performed a separate experiment to confirm it. All methodology and discussion of this experiment along with figure 13 is present in revised version.

10. How does the current material compare to other materials for waste water treatment available in the literature in terms of adsorption capacity? Kindly tabulate for a better comparison.

A table 4 is provided in revised manuscript that clearly compare the current material to other materials for waste water treatment available in the literature in terms of adsorption capacity.

11. Polluted water always contains other contaminants. Kindly present the adsorption capacity of the material in the presence of other contaminants.

This experiment was performed in double distilled water as a model case of water pollution by dyes, and there is no other contaminates. 

12. Kindly define the RDS abbreviation the first time it is used in the manuscript.

It has added in page 2 line 59.

“The raw date seeds (RDSs), in particular, have gotten much press because of their low cost, natural availability, and lack of environmental impact”

13. The manuscript uses terms like “cooked” and “baked”. Kindly consider using scientific terms to describe the methodology.

“cooked” and “baked” terms were replaced with more scientific term as “heated”

14. Kindly have the manuscript undergo language editing to correct various errors, as well as inconsistencies in tenses and writing point of view.

We thoroughly checked the language of the manuscript and run it Grammarly software to further remove the various errors, as well as inconsistencies in tenses and writing.

We have also made some other changes in manuscript, e.g., formatting style, references according to your journal requirements.

We appreciate earnestly for Editors/Reviewers’ warm work and hope that the correction will meet with approval criteria.

Thank you very much again for your comments and suggestions.

Yours sincerely,

Dr. Khalid Ali Khan

On behalf of all authors.

Reviewer 3 Report

The manuscript needs major revisions. The questions and suggestions are listed below.

1.     The title is suggested to be revised as “Characterization of date palm seed powder derived porous carbon/GO composite and its application as an environmental functional material to remove dye from aqueous solutions” since date palm seed powder is carbonized and GO is used. The statements of data palm seed powder should be revised correspondingly.

2.     The writing of date seeds should be in accordance with each other. The preparation process of materials is suggested to be mentioned in the abstract.

3.     Wastewater treatment is important for the sustainable development. Various materials have been developed to remove the dyes, heavy metal ions, and other contaminants. More references are suggested to be cited for broad readers, for example “Synthesis of lignin-poly(N-methylaniline)-reduced graphene oxide hydrogel for organic dye and lead ions removal; Synthesis and Application of Granular Activated Carbon from Biomass Waste Materials for Water Treatment: A Review”.

4.     “cooled the liquid to 0” in line 131 should be revised as “cooled the liquid to 0 oC”.

5.     For “After centrifugation and sonication, the GO solution was rinsed.” in line 137, how does Go solution be rinsed? Are the authors mean dialysis?

6.     The specific surface area of the carbon composites is suggested to be tested since specific surface area is an important issue need to be considered for absorbent. Please refer and cite Inorganic Chemistry Frontiers 2022, DOI: 10.1039/D2QI01914K; Applied Surface Science 2022, 608, 155144.

7.     PTZS is labeled in the Figure 4 while the caption does not mention PTZS. Please double checked it.

8.     The dots in Figure 9 a and c should not be fitted by a red line. They do not show a linear relation. So does Figure 11.

9.     The authors should pay attention to the writing of units. There should be a space between the number and the unit. The units should be written in the same style. For example, “g L-1” and “mg/g” are different styles.

10.  Please pay attention to the references. The writing of journal name should be in the same style.

11.  Please pay attention to the writing of subscripts and superscripts.

Author Response

Ms. No. materials-1990728

Title of the article:

Characterization of date palm seed powder and its application as an environmental functional material to remove dye from aqueous solutions.

RESPONSE LETTER

Reviewer #3:

The manuscript needs major revisions. The questions and suggestions are listed below.

COMMENT

RESPONSE

1. The title is suggested to be revised as “Characterization of date palm seed powder derived porous carbon/GO composite and its application as an environmental functional material to remove dye from aqueous solutions” since date palm seed powder is carbonized and GO is used. The statements of data palm seed powder should be revised correspondingly.

Thank you so much for your suggestion to change the title, we have modified the title almost per your suggestion.

2. The writing of date seeds should be in accordance with each other. The preparation process of materials is suggested to be mentioned in the abstract.

The writing of date seeds is now in accordance with each other through the text. While, the preparation process of materials has been added in the abstract

3. Wastewater treatment is important for the sustainable development. Various materials have been developed to remove the dyes, heavy metal ions, and other contaminants. More references are suggested to be cited for broad readers, for example “Synthesis of lignin-poly(N-methylaniline)-reduced graphene oxide hydrogel for organic dye and lead ions removal; Synthesis and Application of Granular Activated Carbon from Biomass Waste Materials for Water Treatment: A Review”.

We appreciate the valuable comments. Two references have been added as per suggestion for broad readership.

4. “cooled the liquid to 0” in line 131 should be revised as “cooled the liquid to 0 C”.

The statement has been revised accordingly in the revised manuscript.

5. For “After centrifugation and sonication, the GO solution was rinsed.” in line 137, how does Go solution be rinsed? Are the authors mean dialysis?

In mentioned line authors mean GO solution to be rinsed.

6. The specific surface area of the carbon composites is suggested to be tested since specific surface area is an important issue need to be considered for absorbent. Please refer and cite Inorganic Chemistry Frontiers 2022, DOI: 10.1039/D2QI01914K; Applied Surface Science 2022, 608, 155144.

The suggested specific surface area of the carbon composites is tested and cited accordingly in revised manuscript. A new figure 6 is also incorporated. 

7. PTZS is labeled in the Figure 4 while the caption does not mention PTZS. Please double checked it.

It is corrected accordingly.

8.  The dots in Figure 9 a and c should not be fitted by a red line. They do not show a linear relation. So does Figure 11.

We revised all figures and the results obtained from Origin Pro 9 software are added in revise manuscript.

9. The authors should pay attention to the writing of units. There should be a space between the number and the unit. The units should be written in the same style. For example, “g L-1” and “mg/g” are different styles.

It has been corrected in revised manuscript throughout the revised text.

10. Please pay attention to the references. The writing of journal name should be in the same style.

All references are changed according to the journal style.

11. Please pay attention to the writing of subscripts and superscripts.

The manuscript thoroughly checked for correct writing of subscripts and superscripts

We have also made some other changes in manuscript, e.g., formatting style, references according to your journal requirements.

We appreciate earnestly for Editors/Reviewers’ warm work and hope that the correction will meet with approval criteria.

Thank you very much again for your comments and suggestions.

Yours sincerely,

Dr. Khalid Ali Khan

On behalf of all authors.

Round 2

Reviewer 2 Report

The authors prepared an adsorbent from date seeds for dye adsorption. However, the data are not presented clearly and show erroneous data, that is, the statements differ from what is seen in the data figures. The study lacks proper characterization and incomplete data is presented. Application wise, the experiments carried out are lacking to provide sufficient evidence that the material is useful for waste water treatment.

 Comments:

1.       What is the “Equilibrium of adsorption X-ray diffraction” mentioned in the abstract?

2.       The XRD pattern presented in Figure 2 is of poor quality with very low signal-to-noise ratio. Kindly replace with a better measurement and also place the calculated value of graphene oxide and graphite for a better comparison.

3.       Mentioned in page 6 “Fig. 4 displays GO-date Seeds, graphite and graphene oxide SEM images.” However, the images were not clearly labeled in the figure (which is graphite? Graphene oxide?). What is the PTZS label in Figure 4? Also, provide similar magnification of samples for better comparison.

4.       Mentioned in page 6 “The adsorption rate of the GO-date Seeds and GO was investigated using an FTIR spectrometer.” How was the adsorption rate measured by FTIR?

5.       Please improve labels on Figure 5. Include functional groups represented by the peaks so it is easy for readers to interpret in one look.

6.       Kindly present the N2 adsorption data and the pore size distribution mentioned in Page 7.

7.       The dye is acidic in nature and placing it in a basic media gives a tendency for the dye to be unstable and breakdown. Kindly present the blank determination or stability tests whereby the dye is exposed to varying pH and subsequently measured for UV-Vis spectra.

8.       Mentioned in page 9 “Furthermore, the adsorption capacity (Ads. per cent) increased as the initial dye concentration was raised…” However, it can be seen from Figure 8 that the adsorption percent lowers as the initial concentration was raised.

9.       Is the adsorption process reversible? Is the material reusable?

10.   How does the current material compare to other materials for waste water treatment available in the literature in terms of adsorption capacity? Kindly tabulate for a better comparison.

11.   Polluted water always contains other contaminants. Kindly present the adsorption capacity of the material in the presence of other contaminants.

Other comments:

1.       Kindly define the RDS abbreviation the first time it is used in the manuscript.

2.       The manuscript uses terms like “cooked” and “baked”. Kindly consider using scientific terms to describe the methodology.

3.       Kindly have the manuscript undergo language editing to correct various errors, as well as inconsistencies in tenses and writing point of view.

Author Response

Ms. No. materials-1990728

Title of the article:

Characterization of date palm seed powder and its application as an environmental functional material to remove dye from aqueous solutions.

RESPONSE LETTER

Reviewer #2:

The authors prepared an adsorbent from date seeds for dye adsorption. However, the data are not presented clearly and show erroneous data, that is, the statements differ from what is seen in the data figures. The study lacks proper characterization and incomplete data is presented. Application wise, the experiments carried out are lacking to provide sufficient evidence that the material is useful for waste water treatment.

COMMENT

RESPONSE

1. What is the “Equilibrium of adsorption X-ray diffraction” mentioned in the abstract?

We deleted the “Equilibrium of adsorption X-ray diffraction” from the abstract and replaced with X-ray diffraction (XRD), scanning electron microscopy (SEM), and Fourier-transform infrared spectroscopy (FTIR).

2. The XRD pattern presented in Figure 2 is of poor quality with very low signal-to-noise ratio. Kindly replace with a better measurement and also place the calculated value of graphene oxide and graphite for a better comparison.

The XRD pattern was generated through software Origin 95, so based on our data file it is difficult to replace. In this experiment we worked on GO-date seeds and peaks for graphite was used only for comparison. 

3. Mentioned in page 6 “Fig. 4 displays GO-date Seeds, graphite and graphene oxide SEM images.” However, the images were not clearly labeled in the figure (which is graphite? Graphene oxide?). What is the PTZS label in Figure 4? Also, provide similar magnification of samples for better comparison.

The revised figure has only GO-date seeds before and after adsorption experiment. Further, the label inside figure 4 is changed to “before” and “after”

4. Mentioned in page 6 “The adsorption rate of the GO-date Seeds and GO was investigated using an FTIR spectrometer.” How was the adsorption rate measured by FTIR?

The adsorption rate was not measured by FTIR? But the functional groups were measured by FTIR. So, we changed the phrase “The adsorption rate” with “functional groups”

5. Please improve labels on Figure 5. Include functional groups represented by the peaks so it is easy for readers to interpret in one look.

Figure 5 improved and revised according to the suggestion.

6. Kindly present the N2 adsorption data and the pore size distribution mentioned in Page 7.

The N2 adsorption data were improved as mentioned from line number 279-287.

7. The dye is acidic in nature and placing it in a basic media gives a tendency for the dye to be unstable and breakdown. Kindly present the blank determination or stability tests whereby the dye is exposed to varying pH and subsequently measured for UV-Vis spectra.

The dye was stable when placing in nature and a basic media. Whereas, the absorbance was semi-identical as shown in the figure S4 in supplementary file.

8.  Mentioned in page 9 “Furthermore, the adsorption capacity (Ads. per cent) increased as the initial dye concentration was raised…” However, it can be seen from Figure 8 that the adsorption percent lowers as the initial concentration was raised.

Thank you for your value comment. We have corrected the text in revised version as mentioned at page 12 (L332) which is describing the figure 9 (before Fig.8) properly.

9. Is the adsorption process reversible? Is the material reusable?

Yes, the adsorption process is reversible and the material is reusable. We performed a separate experiment to confirm it. All methodology and discussion of this experiment along with figure 13 is present in revised version as new section in results part under 3.7. The reusable of GO-date seeds in adsorption (Page 20; L472-L477).

10. How does the current material compare to other materials for waste water treatment available in the literature in terms of adsorption capacity? Kindly tabulate for a better comparison.

A table 4 is provided in revised manuscript that clearly compare the current material to other materials for waste water treatment available in the literature in terms of adsorption capacity (Page 19; L468-L469).

11. Polluted water always contains other contaminants. Kindly present the adsorption capacity of the material in the presence of other contaminants.

This experiment was performed in double distilled water as a model case of water pollution by dyes, and there is no other contaminates. As mentioned at page 5; L164. 

12. Kindly define the RDS abbreviation the first time it is used in the manuscript.

It has added in page 2 line 64.

“The raw date seeds (RDSs), in particular, have gotten much press because of their low cost, natural availability, and lack of environmental impact”

13. The manuscript uses terms like “cooked” and “baked”. Kindly consider using scientific terms to describe the methodology.

“cooked” and “baked” terms were replaced with more scientific term as “heated”

Page 4; L-131; L141. 

14. Kindly have the manuscript undergo language editing to correct various errors, as well as inconsistencies in tenses and writing point of view.

We thoroughly checked the language of the manuscript and run it Grammarly software to further remove the various errors, as well as inconsistencies in tenses and writing.

We have also made some other changes in manuscript, e.g., formatting style, references according to your journal requirements.

We appreciate earnestly for Editors/Reviewers’ warm work and hope that the correction will meet with approval criteria.

Thank you very much again for your comments and suggestions.

Yours sincerely,

Dr. Khalid Ali Khan

On behalf of all authors.

Reviewer 3 Report

The manuscript has been revised according to the comments and is much better than the former version. But further revisions are needed before acception. Suggestions and questions are listed below.

1. Pore size distribution curves are suggested to be added in Figure 6. Please refer and cite Nanoscale 2022, 14, 8216; Diamond and Related Materials 2022, 130, 109465.

2. Figure 10 a and c should not be fitted by a straight line. The data do not show a linearship.

3. Please check 103 in table 1. Is it 103 or 10^3 (1000)?

4. Figure 12 is not clear and the data in b and d should not be fitted by a straight line.

5. Please check the writing of subscript for chemicals in references.

Author Response

Ms. No. materials-1990728

Title of the article:

Characterization of date palm seed powder and its application as an environmental functional material to remove dye from aqueous solutions.

RESPONSE LETTER

Reviewer #3:

The manuscript has been revised according to the comments and is much better than the former version. But further revisions are needed before acceptation. Suggestions and questions are listed below.

COMMENT

RESPONSE

1. Pore size distribution curves are suggested to be added in Figure 6. Please refer and cite Nanoscale 2022, 14, 8216; Diamond and Related Materials 2022, 130, 109465.

The pore size distribution curve was added in the figure 6 (page 10; line 288). Furthermore, Nanoscale 2022, 14, 8216; & Diamond and Related Materials 2022, 130, 109465 were cited in text as 43 and 44 at page 10; line 287.

2. Figure 10 a and c should not be fitted by a straight line. The data do not show a linearship

Thank you for valuable comment. Yes, the data are not fitted to straight line and these results were explained in text (Page 15; L-368 to L-376)

3. Please check 103 in table 1. Is it 103 or 10^3 (1000)?

It is corrected as103 instead of 103 in Table-1; Page 16; Line-383

4. Figure 12 is not clear and the data in b and d should not be fitted by a straight line.

Thank you for valuable comment. Yes, the data are not fitted to straight line and these results were explained in text (Page 16; L-395 to L-413)

5. Please check the writing of subscript for chemicals in references.

We carefully checked the whole manuscript and corrected accordingly.

We have also made some other changes in manuscript, e.g., formatting style, references according to your journal requirements.

We appreciate earnestly for Editors/Reviewers’ warm work and hope that the correction will meet with approval criteria.

Thank you very much again for your comments and suggestions.

Yours sincerely,

Dr. Khalid Ali Khan

On behalf of all authors.